# Whose centre is it anyway? Defining person-centred care in nursing: An integrative review

**Amy-Louise Byrne**◯*, **Adele Baldwin, Clare Harvey**◯

Central Queensland University School of Nursing, Midwifery and Social Science, Townsville, Queensland, Australia

* a.byrne@cqu.edu.au

## Abstract

### Aim

The aims of this literature review were to better understand the current literature about person-centred care (PCC) and identify a clear definition of the term PCC relevant to nursing practice.

### Method/Data sources

An integrative literature review was undertaken using The Cumulative Index to Nursing and Allied Health Literature (CINAHL), Medline, Scopus and Pubmed databases. The limitations were English language, full text articles published between 1998 and 2018 within Australian, New Zealand, Canada, USA, Europe, Ireland and UK were included. The international context off PCC is then specifically related to the Australian context.

### Review methods

The review adopted a thematic analysis to categorise and summarise themes with reference to the concept of PCC. The review process also adhered to the Preferred Reporting System for Meta-Analysis (PRISMA) and applied the Critical Appraisal Skills Programme (CASP) tools to ensure the quality of the papers included for deeper analysis.

### Results

While definitions of PCC do exist, there is no universally used definition within the nursing profession. This review has found three core themes which contribute to how PCC is understood and practiced, these are *People*, *Practice and Power*. This review uncovered a malalignment between the concept of PCC and the operationalisation of the term; this misalignment was discovered at both the practice level, and at the micro, meso and micro levels of the healthcare service.

### Conclusion

The concept of PCC is well known to nurses, yet ill-defined and operationalised into practice. PCC is potentially hindered by its apparent rhetorical nature, and further investigation of how PCC is valued and operationalised through its measurement and reported outcomes is

**Data Availability Statement:** All relevant data are within the manuscript and its Supporting Information files.

**Funding:** The author(s) received no specific funding for this work. The publication is funded

under the first authors Research Higher Degree (PhD) budget.

**Competing interests:** The authors have declared that no competing interests exist.

needed. Investigation of the literature found many definitions of PCC, but no one universally accepted and used definition. Subsequently, PCC remains conceptional in nature, leading to disparity between how it is interpreted and operationalised within the healthcare system and within nursing services.

## Introduction

Healthcare is changing, for both providers and recipients of care, with ongoing challenges to traditional roles and power balances. The causative factors of changes to the way healthcare is provided are complex, but one contributing factor is easier access to healthcare information and better-informed populations [1] whereby people as healthcare consumers have access to healthcare information through multiple media. On the surface, consumers are no longer seen as passive recipients of care, but rather as valuable and active members of the healthcare team. The concept of Person-Centred Care (PCC) is used to describe a certain model for the role of the patient within the healthcare system and the way in which care is provided to the patient [2,3]. Globally, there is continued advocacy for person-centred, individualised care [4], with the contemporary term for PCC being frequently presented in healthcare discourse, and frequently associated with the safety and quality of healthcare service provision [5,6]. Indeed, partnering with consumers within a person-centred framework is now a fundamental requirement for Australian healthcare services, meaning that they cannot achieve accreditation without demonstration of PCC [7]. Hence, PCC is now seen in healthcare service strategy and models of care, designed to support the voice of the patient and the role of the healthcare service in engaging with patients [6]. PCC also forms part of the Australian nursing professional standards [8] yet is paradoxically described as an 'extra' to nursing practice [9], taking a back seat to nursing tasks and errands that make up the day to day regime of the nurse.

Despite the discourse around PCC, and the requirements of PCC within healthcare, there appears to be no universally accepted definition of the term. This leaves the concept open to interpretation and potential confusion, particularly when personnel, in this case nurses, attempt to operationalise it. This review investigated the meaning of PCC with reference to nurses across different practice settings and specialities. To further facilitate the understanding [10] and theory development of the concept of PCC, this review adopted an integrative review methodology [11].

## Background

In the late 1950's and 60's, PCC, and care for the entire self was first described in the context of psychiatry, such as in Rogers' 'On becoming a person' [12]. Patient-centred medicine was a term first coined by psychoanalyst Michael Balint. Balint was instrumental in the education of general practitioners around psychodynamic factors of patients and challenged the traditional illness-orientated model [13]. Balint's challenge extended beyond the traditional healthcare model to include both the physical and psychosocial as part of the practitioner's role. Balint explained; *"Here, in additional to trying to discover a localized illness or illnesses, the doctor also has to examine the whole person in order to form what we call an 'overall diagnosis.' The patient, in fact, has to be understood as a unique human-being."* [13 p269].

The idea of caring for the whole person, and the divide between traditional medical practice and the psychosocial needs of the patient was discussed by Engle in 1977. He wrote; *"The dominant model of disease today is biomedical. . .It assumes disease to be fully accounted for by*

*deviations from the norm of measurable biological (somatic) variables. It leaves no room within its framework for the social, psychological, and behavioural dimensions of illness.*" [14 p379]. The biopsychosocial model proposed provided a new basis for care which included care of the mind and body. Over the succeeding years, this model of care and the notion of patient centre care continued to evolve, with many iterations of the term moving with the changing climate of healthcare systems.

PCC gained significant traction through the Institute of Medicines (IOM) 2001 report 'Crossing the Quality Chasm: A New Health System in the 21st Century' [15] as a key element of quality healthcare. The IOM provided one of the first contemporary definitions, stating that PCC "*encompasses qualities of compassion, empathy and responsiveness to the needs, values and expressed preferences of the individual patient*" [15 p48]. The World Health Organization continues to advocate for integrated care that is in tune with the patient's wants and needs through the framework on Integrated People-Centred Care. This includes the vision that "*all people have equal access to quality health services that are co-produced in a way that meets their life course needs*". [5] This framework aims to improve engagement of people and communities, strengthen governance and accountability, reorientate the model of healthcare, and coordinate services across sectors [5], seeing people as important contributors and decision makers over their own care.

More recently, the Australian Commission on Safety and Quality in Health Care (ACSQHC) defines PCC as an "*innovative approach to the planning, delivery, and evaluation of health care . . .[involving] mutually beneficial partnerships among health care providers, patients, and families.*" [2 p13]. Thus, PCC has become an integral element of care from a quality, planning and practice level, and therefore appears prominently in Australian healthcare service discourse and associated models of care, often presented as an underpinning philosophy for the way in which nursing care is provided [3]. The concept of PCC continues to evolve, notably in the change to 'person' rather than 'patient' in recognition of the whole person, not simply the disease process. Other variables such as Family-centred care are used more in the context of aged care and paediatrics [16,17].

As the term has become more common in healthcare discourse, frameworks have emerged to allow the term to be operationalised into practice. There are several person-centred nursing frameworks including the Senses [18], VIP [19], 6 C's [20], The Burford Model [21] and McCormack and McCance's framework [22]. These frameworks describe elements such as attributes of staff, methods of interactions, coordination of care and services, the care environment and consideration of outcomes of care. These examples provide insight into attempts to operationalise PCC, into individual practice and healthcare service provision.

Nurses are the healthcare professionals who spend the most time with people and are therefore in a position to act as their advocates, with nursing staff managing the continuity of care [23]. This review seeks to investigate the meaning of person-centred nursing practice, and acts as a starting point for a wider study into the concept of PCC for people with long term conditions. Consumers of healthcare navigate a complex and fragmented system, with fragmentation leading to patients feeling lost, and a decrease in the quality of services offered [24]. This places even greater importance on a partnership between provider and receiver, particularly in the face of increasing chronicity/complexity of care. Within this fragmented and complex system, the patient must always remain at the centre of their care. Hence, there is a need for a robust definition to ensure PCC is more clearly operationalised and care delivered is designed around the needs of the patient, rather than trying to make the patient fit within the system.

This review uses the term person rather than patient in recognition of the person as a whole. Where clarity is required, the term healthcare consumer is used; a term frequently used in Australia.

## Aim

The aim of this literature review was to understand better from the literature how nurses operationalise the definition of PCC.

## Search questions

This literature review sought to answer the following questions:

- Is there a commonly/generally accepted definition for PCC that is used by nurses?

- How do nurses operationalise PCC in practice?

## Search strategy

An integrative literature review was conducted using the terms Person Cent* Care OR Patient Cent* Care AND Nurs* AND Definition OR Meaning OR understanding OR Concept. The search was expanded to include similar terms and concepts such as patient/person-centredness and personalisation. A major subject heading of 'patient centred care' was used within the searches. This review is positioned within the nursing discipline; therefore, articles were included if they were specific to nursing or if they included nursing texts in the review. English language, full text articles published between 1998 and 2018 were included. Publications from Australia and New Zealand, Canada, USA, Europe, UK and Ireland were included to gain an understanding of PCC in the western context. The Cumulative Index to Nursing and Allied Health Literature (CINAHL), Medline, Scopus and Pubmed databases were searched. This search is registered with PROSPERO (ID number 148778) and was completed in March 2019. While the search strategy includes international literature, this will be related back to the Australian context, in order to understand how PCC operates within Australia.

## Data extraction

The framework, from Whittemore and Knafl [11], describes a comprehensive review, identifying the maximum number of eligible primary sources and requires the researcher to explicitly justify decisions made in the sampling. Using this framework, a total of 1817 articles met the search terms, highlighting the volume of literature available on the concept of PCC. Table 1 provides the scope of inclusions and exclusions. From this, 255 articles were selected for review.

**Table 1. Inclusion and exclusion criteria.**

| Inclusions | Exclusions |
|---|---|
| Articles related to Nursing or included nursing | Articles relating to Midwifery and 'women-centred care' |
| All nursing specialty areas including stroke, Intensive care, aged care, acute care, operating theatres chronic disease | Concept analysis for concepts such as 'self' and 'cultural competence', 'compassion' or 'empowerment' where these were discussed in isolation |
| Staff and patient perceptions | Trial registrations and study protocols |
| Person-centred care frameworks | Professional role development and leadership |
| Family centred care | Digital health and integrated technology |
|  | Articles where PCC was not major subject; for example, aggression and hand hygiene |

After removal of duplicates, 203 articles were subjected to full review. A further refined strategy excluded Key Performance Indicators (KPI), service measures, assessment tools and validations as the goal was defining the term, rather than to assess how it is measured; these represented a large proportion of the articles within the search. A total of 44 articles were subjected to quality review. To ensure adequate rigour, reliability and relevance, all articles were evaluated against the Critical Appraisal Skills Programme (CASP) systematic and qualitative review checklists [25,26] by the lead author and reviewed by a senior researcher on the team. The relevance of the papers and the quality of the reviews/articles themselves was appraised. All articles were appraised against the aims of this review. Following this, a total of 17 articles were included in the final review. Fig 1 provides the summary for the search process while S1 File provides the PRISMA checklist.

Using the previously identified framework that allowed for data from diverse methods and approaches to be analysed and compared, a constant comparison method was used to convert data from different categories into patterns, themes and relationships. The data is thus displayed below in Table 2 to encompass the full depth of the concept and to provide new understanding, and its implications to practice [11]. Table 2 demonstrates the characteristics of the articles reviewed including their design methods, populations and findings.

## Findings

This review set out to investigate if a universal definition of PCC for nursing exists and is used; what was uncovered was a deeper understanding of the concept and operationalisation of PCC, highlighting a malalignment between concept and reality. Three (3) core themes were identified in the review process, each of which is comprised of two (2) sub-themes. These three core themes of *People*, *Practice*, *and Power*, with the respective sub-themes are discussed are summarised in Table 3.

### Theme 1: People

Unsurprisingly, the most common threads in the literature about PCC relate to people and, consistent with the philosophy of PCC, is described as basic, human kindness and respectful behaviour [22,27,28,29]. The core theme of People comprises two sub-themes: *Recognising uniqueness* and *Partnerships*.

**Recognising uniqueness.** PCC, as the name suggests, is care that is considered and based on the individual person, who is the recipient of care. Prominent in the literature are the concepts of personhood, individuality and uniqueness [16,28,30]. Individuality, and the sense of self, understands that each person has their own unique wants, needs and desires [16,29] Personhood reinforces and values the complete person, with an understanding that illness affects the entire person [31]; an holistic consideration of the person that extends to family interventions and involvement [27,32] described as developing and maintaining trust within the family unit [33,34]. Uniqueness is central to this subtheme as recognition of the person as a unique being leads to unique and tailored care, based on the needs of the whole person [16,29].

**Partnership.** The literature discusses the need for a relationship between healthcare provider and healthcare receiver as a way of facilitating information, knowledge and decision making. The term 'relationship' is prominent in the literature including the terms therapeutic relationship [16], clinical relationship [35] and partnership [29,36,37]. This is described in the contexts of cohesive, cooperative teams [29,32], mutuality between provider and receiver [38], and the balance of power and the sharing of knowledge [16]. These themes are further developed through the practice of the nurse and are thus carried forward to the next theme, Practice.

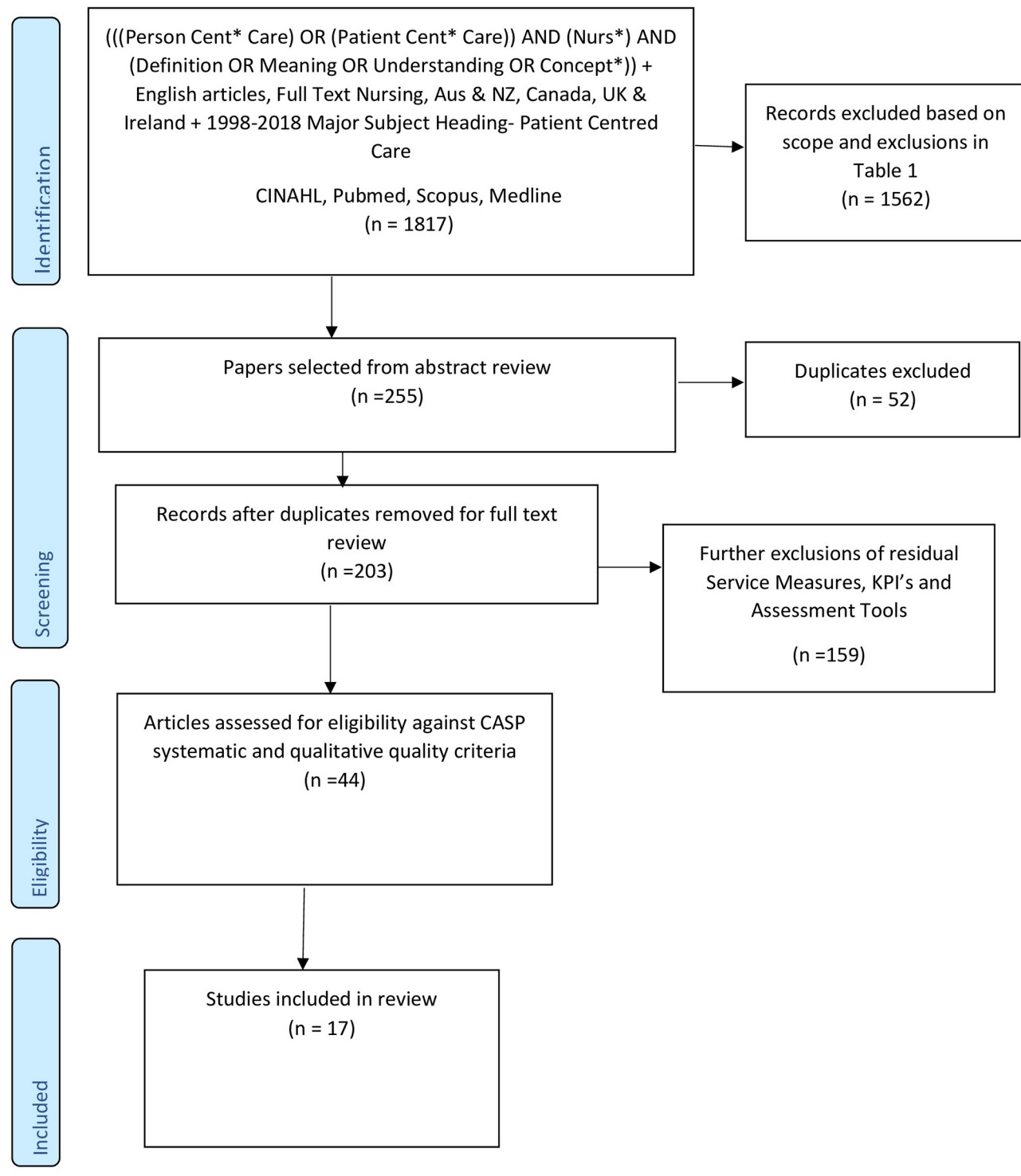

**Fig 1. Literature review search process.**

**Table 2. Literature review findings.**

| Authors, year, location | Population | Design methods | Findings/Comments | Common authors and articles |
|---|---|---|---|---|
| **LITERATURE REVIEWS** | | | | |
| Slater, L (2006) AUS | Search of term Person/patient-centeredness personhood or related words CINAHL, Ovid, Science direct, Medline, Blackwell Synergy and Wiley Interscience | Systematic Review and Concept Analysis | Recognition of Personhood; Individuality, Self, Rational Decision Making, Reflection on available choices, Needs, wants and desires Evidence of a therapeutic relationship between person and health provider; Mutuality, Decisions are valued, power is balanced, therapeutic relationship Respect for Individuality of the person Uniqueness, Individuality, 'person' not 'patient', own values, autonomy, cultural needs Provision of care that reflects professional ethical standards Autonomy, dignity, privacy, rights Identification and reinforcement of the person's strengths and positive aspects rather than the weaknesses and problems Valued, wellness v's Illness, the person as a whole Acknowledgement of the person's lived world Lived experiences, current experiences, person's story Empowerment of the person to make their own decisions about their health Shift of power, Providing knowledge, ability to make decisions, respect for final decisions Definition: Not provided; Advocates strongly for the term 'Person' rather than 'Patient or 'Client' as a salute to not only the person's right to care but to have choices as to how it is perceived and provided. | • Downs (1997) the emergence of the Person in dementia research • Ford & McCormack (2000) keeping the person in centre of nursing • Kitson (1999) The essence of nursing (part I and II) • Kitwood (1997) the experience of Dementia • McCormack (2003) A Conceptual framework for person-centred practice with older people • Nolan, Keady, Aveyard (2001) Relationship-centred care is the next logical step • Price (2004) demonstrating respect for patient dignity • Stewart (2001) towards a global Definition of patient-centred care |
| McCormack, McCance (2006) Ireland | Developed from 14 case studies of nurse-patient relationships and other work by authors (used to develop framework) Acute care setting, though authors have considerable work in aged-care | Systematic iterative process | Prerequisites Attributes of nurses, professional competence, developed interpersonal skills, being committed to the job, clarity in belief, values and knowing self and insight The Care Environment Supportive organisation systems, sharing of power and potential for innovation and risk, workplace culture, quality of leadership, thorough evaluation Person-centred Process Having a sympathetic presence, sharing decision making, engagement, working with the patient's beliefs and values, providing for physical needs Definition: Not provided- provides a framework for PCC in practice which hence forms meaning in nursing practice | Influential authors with multiple papers leading to this framework • McCormack (2001) Negotiating partnerships with older people- A person-centred approach • McCormack (2003) A conceptual frame work for person-centred practice with older people • McCormack (2004) Person-centeredness in Gerontological nursing: an overview of the literature • McCance (2003) Caring in nursing practice: the development of a conceptual framework • Mead & Bower (2000) Patient Centeredness: a conceptual framework and Review of the empirical literature • Nolan, Davis, Brown, Keady & Nolan (2004) beyond person-centred care: a new vision for gerontological nursing |
| Lawrence, Kinn (2012) UK | Review of PCC in stroke literature Medline and Psychinfo, AMED, ASSIA, BNI, Cochrane, DARE, CCTR, CINAHL, Embase between 1994–2010 21 studies | Systematic mixed methods | Meaningfulness and relevance Understanding patient experience Ascertaining priorities, concerns and patient goals Quality of experience Measuring person centred care support The need to understand the experience of caregivers Family centred interventions Quality participation activities Communication Communication with impaired adults and families Definition: PCC- 'Identifies individual's communication skills and utilises appropriate and effective communication strategies in all interactions between the health-care professional and the individual. Identifies outcomes that are valued and prioritized by individuals. Identifies outcomes that reflect the desire quality of participation. Monitors and measures outcomes at appropriate times and points om the rehabilitation process. Uses the s resultant information to inform the patient/health-care professional's decision-making process. | • Mead & Bower (2000) Patient-centeredness: a conceptual framework and review of the empirical literature • Kennedy (2003) Patients are experts in their own field • Gillespie Florin, Gillam (2004) How is Patient-centred care understood by clinical, managerial and lay stakeholders responsible for promoting this agenda • Mead & Bower (2002) Patient-centred Consultations and outcomes in primary care • Lawrence (2009) Patient-centred stroke care: young adults and their families |

*(Continued)*

**Table 2.** (Continued)

| Authors, year, location | Population | Design methods | Findings/Comments | Common authors and articles |
|---|---|---|---|---|
| Kitson, Marshall, Bassett, Zeitz (2013) AUS | Key and seminal texts from Nursing and Medical looking at patient-centred care search of CINAHL, Scopus, Medline between 1990–2010. 60 papers included | Narrative review and Concept Analysis | Patient participation and involvement; Patient participating as a respected and autonomous individuals- respect for patients, values preferences, expressed needs, patient as a source of control, patient actively involved and participation, autonomy Care Plan based on patient needs- Care customised, according to needs and values, transition to community Addressing a patient physical and emotional needs- physical comfort, physical care, emotional support, alleviation of anxiety Relationships between health professionals; Genuine clinician relationship, open communication, knowledge, clinical expertise v patient experience, health professional skills and knowledge, cohesive and co-operative team professional The context of care- core systems; Policy and practice, continuum language, access, barriers to PCC, supportive organisational systems, therapeutic environment Definition: Not provided: Though found consistent themes between nursing and medicine. Conclude that there are core elements that transcend professional boundaries, though different professions place greater importance on certain elements. Nursing tends to accentuate respect for patient values and beliefs | • Balint (1969) the possibilities of patient centred medicine<br>• Edvardsson, Nay (2010) cute care for Older: challenges and ways forward<br>• Epstein (2000) the science of person-centred care<br>• Institute of Medicine (2001) Crossing The quality chasm<br>• Kitson (2002–2010) recognising Relationships; The need for system changes; Defining the fundamentals of care<br>• Marshall, Kitson & Zeitz (2012) patient views on patient-centred care<br>• McCance, Slater, McCormack (2009) Using the caring dimensions of inventory as an indicator of person-centred nursing<br>• McCormack (2003–2004) A conceptual Framework for person-centred practice nursing; Researching nursing practice: does person-centredness matter? Person-Centeredness in gerontological nursing<br>• McCormack & McCance (2006) Development of a framework for person-centred nursing<br>• Mead & Bower (2000) Patient-Centeredness: a conceptual framework and review of the empirical literature<br>• Nolan, Davis, Brown, Keady, Nolan (2004) beyond person-centred care: a new vision for gerontological nursing<br>• Price (2006) exploring person-centered Care<br>• Stewart (2001) toward a global Definition of patient centred care<br>• Zeitz, Kitson et al (2011) working together to improve the care of older persons |
| Morgan, Yoder (2012) USA | Concept analysis of PC, CINAHL, Medline, Pubmed and Cochrane review 50 articles included | Systematic concept review | Holistic Recognises and values the whole person Illness affects the whole person Individualised Understanding the person's life situation and their abilities Decision making and control Respectful The 'right' thing Recognising the individual as competent to decide Offering choice Empowering Promotes self-confidence and self determination Definition: Not provided however advocates for providing clarity in the concept as a way of improving PCC in acute care. | • Balint (1968) The possibilities of Patient-centered medicine<br>• Douglas & Douglas (2005) Patient-Centered improvements in healthcare-built environments<br>• Edvardsson, Koch & Nay (2009) Psychometric evaluation of the English language person-centered climate questionnaire- patient version<br>• Hobbs (2009) A dimensional analysis of patient-centered care<br>• Institute of Medicine (2001) Crossing the quality chasm<br>• Kitson (1986) Indicators for quality in Nursing care- an alternative approach<br>• McCance (2003) caring in nursing Practice: the development of a conceptual framework<br>• McCormack and McCance (2003) A Conceptual framework for person centred practice with older people<br>• Mead & Bower (2000) Patient-centeredness: a conceptual framework and review of the empirical literature<br>• Rogers (1961) on becoming a person<br>• Slater (2006) Person-centredness: A concept analysis<br>• Stewart, Brown, Weston, McWhinney, McWilliam, Freeman (1995) patient-centered medicine: transforming the clinical method |

(*Continued*)

**Table 2.** (*Continued*)

| Authors, year, location | Population | Design methods | Findings/Comments | Common authors and articles |
|---|---|---|---|---|
| Lusk, Fater (2013) USA | CINAHL, PsychInfo and Medline for search of combination of terms Patient-centred, power and research 2001–2010 extended by hand search. 24 articles included | Review and Concept Analysis | Attributes of healthcare providers; caring, moral and ethical behaviour, faith and hope, sensitivity, trust, relationships, teaching and learning, listening Autonomy; The right to make decisions, creative problem solving, individuality The lived experience; Understanding and working within the lived experience Power; Decision making, individualised care, shared decision making, unique individual Outcomes Biophysical markers, physical and social health, access to care and care coordination, care and costs Definition; The provision of care incorporating contextual elements and including the attributes of encouraging patient autonomy, the caring attitude of the nurse, and individualising patient care by the nurse. Behaviours fundamental to the provision of PCC include communicating and listening, treating the patient as a unique individual, respecting values and responding to patient needs. | • Epstein, Fiscella, Lesser, Stange (2010) Why the nation needs a policy push on patient-centered health care • Hobbs (2009) A dimensional analysis of patient-centered care • Institute of medicine (2003) crossing the quality chasm: a new health system for the 21st century • McCormack, Karlsson, Dewing, Lerdal (2010) exploring person-centeredness |
| Jakimowicz, Perry (2015) AUS | Reference to intensive care nursing. CINAHL, Psychinfo, Medline, Pubmed between 2000–2014 28 articles included | Systematic review concept analysis | Biomedical Nursing Routines, complex care, patient survival, nursing knowledge and expertise, technical aspects Patient Identity Maintaining personal identity, understanding vulnerability, treating the patient as unique, fear, lack of control, participation Compassionate Presence Presence while caring, emotional support, allaying fear and anxiety, humanistic, spiritual Professional presence Professional and ethical standards, protect the patient, safety and quality of care, communication, patient advocacy, privacy Definition: Not provided but differentiates the practice of PCC in the technical environment of the ICU. Core themes listed above, however, are consistent with other areas of nursing | • Dewing (2002) from ritual to relationship: a person-centred approach to consent in qualitative research with older people who have dementia • Esmaeili, Cheraghi, Salsali (2014) Critical care nurses understanding of the concept of patient-centered care in Iran nurses • Hobbs (2009) A dimensional analysis of patient-centered care • Kitson, Marshall, Bassett, Zeitz (2013) What are the core elements of patient-centered care? • Kitwood (1997) Dementia reconsidered the person comes first • McCormack (2003) a conceptual Framework for person-centered practice with older people • McCormack and McCance (2010) person centered nursing: theory and practice • Mead & Bower (2000) patient Centeredness: a conceptual framework and review of the empirical literature • Nolan, Davis & Grant (2001) working with older people and their families • Rogers (1961) on becoming a person • Stewart (2001) toward a global definition of person-centered care • World Health Organization (2014) Who global strategy on people-centered and integrated health services |
| Castro, Van Regenmortel, Vanhaecht, Sermeus, Hecke (2016) Europe | Systematic review of Pubmed, Web of science, Embase for Patient empowerment' 'patient participation' 'patient centre' and 'patient-centredness' along with 'conceptual definition'. 20 definitions of patient empowerment, 13 of patient participation and 20 of patient centredness were included. | Systematic review and concept analysis | Relationships; Partners, mutuality, family Shared Decision Making; Knowledge sharing, power Attributes; Communication, respect, values, choices, empathetic, respectful, compassionate, non-judgemental Uniqueness; Expectations, individual, needs, values, beliefs Outcomes; Biopsychosocial, holistic, sharing of clinical knowledge Definition: Patient-centredness is a biopsychosocial approach and attitude that aims to deliver care that is respectful, individualized and empowering. It implies the individual participation of the patient and is built on a relationship of mutual trust, sensitivity, empathy and shared knowledge. | • Balint (1964) The doctor his patient and the illness • McCormack & McCance (2006) Development of a framework for person-centred nursing • Mead & Bower (2000) patient-Centeredness: a conceptual framework and review of the empirical literature • Lusk & Fater (2013) A conceptual Analysis of patient-centered care • Bassett, Kitson, Marshall & Zeitz (2013) What are the core elements of patient-centred care? A narrative review and synthesis of the literature • Morgan & Yoder (2012) A concept Analysis of person-centered care • Stewart (2001) towards a global Definition of patient-centred care |

(*Continued*)

**Table 2.** (Continued)

| Authors, year, location | Population | Design methods | Findings/Comments | Common authors and articles |
|---|---|---|---|---|
| Arakelian, Swenne, Lindberg, Rudolfsson, Vogelsang (2016) Europe | Perioperative nurses' perspectives of PCC Systematic review of Pubmed, CINAHL 2004–2014 23 articles included | Integrative review | Being recognised as a unique entity and being allowed to be the person you are<br>Entire body, being an individual with a name, dignity and respect, being seen as unique, getting to know the person tact and discretion, creating conditions to see the person as an individual<br>Being considered important and person wishes taken into account<br>Self-control and dependency, sharing a story and creating a relationship, connecting, listened to, giving the person time and information, asking questions and appreciating personal belief<br>The presence of the nurse is calming and prevents loneliness, promotes wellbeing<br>Ease anxiety, reduce the feeling of being alone, taking care and being close, being welcomed and expected, feeling safe, feeling like an equal<br>Being close to and being touched by the nurse<br>Emotionally and physically present, staying close and touching patient, creating a feeling of security<br>Definition: PCC means 'being respected as a unique person, being showed consideration, tact and discretion and being taken seriously. Being expected and welcomes by the perioperative nurse when arriving. . .leads to a warm and relaxed atmosphere. . . having access to one's own nurse. . . preventing feelings of loneliness.' | • Brooker (2003) what is person-centred care in dementia<br>• Ekman et al (2011) Person-centered care ready for prime time<br>• Eriksson (2007) becoming through suffering-the path to health and holiness<br>• Price (2006) exploring person-centered care |
| Kogan, Wilbur, Mosqueda (2016) USA | Chronic Disease and functional limitation. Review of literature from CINAHL, Medline, Cochrane, Pubmed 1990–2014 132 included | Systematic review and analysis of themes for definition and for measurement tools | Holistic Care<br>Whole person care, respect and value, choice and dignity, self determination<br>Purposeful living<br>Encouragement and continued social roles<br>Coordinated care<br>Integrated, focused and targeted, multidisciplinary, connected physical health and support services<br>Involving family and friends<br>Definition: Not provided, however advocates for a clear definition as a method of guiding PCC practice and measurements. | • World Health Organization (2013) Towards Person-centred health systems: an innovative approach for better health outcomes<br>• Institute of Medicine (2001) Crossing The quality chasm: a new health system for the 21st century<br>• Epp (2003) person-centred dementia Care: a vision to be refined<br>• Li & Porock (2014) Resident outcomes Person-centered care: A narrative review of interventional research<br>• Brooker (2004) What is person-centered care in dementia<br>• McCormack & McCance (2010) Person-centered Nursing: theory and practice<br>• Edvardsson, Winblad, Sandman (2008) Person-centered care for people with Alzheimers<br>• Edvardsson, Sandman, Borell (2014) Implementing national guidelines for person centered care of people with dementia in residential aged care<br>• Epstein, Fiscella, Lesser (2010) Why the nation needs a policy push on patient-entered health care. |
| **QUALITATIVE REVIEWS** | | | | |
| Edvardsson, Fetherstonhaugh, Nay (2010) AUS | Interviews with aged care staff, people with early onset dementia and family members of patients with dementia N = 67 | Interview and thematic analysis | Promoting a continuation of self<br>Being the person you are and supporting people to continue this<br>Acknowledging the person as valuable, respect, creating opportunities to do likeable things and make decisions<br>Preservation of self<br>Knowing the person<br>Knowing history, preferences, needs and interest and particularities, translating this into practice<br>Welcoming Family<br>Developing and maintain trust, actively communicating, creating opportunities for beneficial teamwork<br>Providing meaning activities<br>Providing meaningful content, self-esteem, creating a feeling of being able to participate, being sensitive<br>Being in a personalised environment<br>Personalising the environment and the care, recognition, person behind the disease<br>Experiencing flexibility and continuity<br>Flexible care and outcomes, adapting care, staff being available, present and willing<br>Definition: Not provided but provides key elements for aged care patients. | |

*(Continued)*

**Table 2.** (Continued)

| Authors, year, location | Population | Design methods | Findings/Comments | Common authors and articles |
|---|---|---|---|---|
| Gachoud, Albert, Kuper, Stroud, Reeves (2012) Canada | General internal medicine. Interviews with nurses, social workers and medical professionals N = 28 | Comparative study, interpretive phenomenological approach | Person centred practice as a philosophy of care<br>Values based and driven, underpinning practice, an element of caring practice, values held by the physician<br>PCP and collaboration<br>Holistic practice, good communication, patient advocacy, patient autonomy, shared decision making, empowerment, quality of care, family involvement and a rapport with the patient<br>Definition: Looks at Person-centred Practice (PCP). Definition not provided but found that nurses and social workers both position themselves as providers of PCC and PCP. Found that Medical Officers were happy to see themselves as lower in the hierarchy of PCP, seeing it as more of a nursing practice | |
| Marshall, Kitson, Zeitz (2012) AUS | Surgical unit from the patient's perspectives N = 10 | Interviews and thematic analysis | Staff<br>Being attentive, being helpful and timely, making an effort, meeting needs and being nice, connectedness, relationships, communication and advocacy<br>Systems<br>Resources, physical environment, workload, senses of loss of control, ward culture empowerment, waiting<br>Definition: Not provided. Finds that PCC model needs to be integrated, incorporating both existing definitions and conceptualisations of PCC which are largely informed by professionals and the meanings and understandings patients give to PCC. | |
| Trajkovski, Schmeid, Vickers, Jackson (2012) AUS | Neonatal Nurses in acute hospital setting N = 33 from nurses' perspective | Focus groups and qualitative interpretive approach | Getting to know parents and their wishes<br>Individualising care based on knowledge of the family unit, trusting relationships, conversations, acknowledgement of each family as different, cultural sensitivity<br>Involving the family in care<br>Share information and guiding families, respect, facilitating the relationship, entire family as decision makers<br>Finding a happy medium<br>Involving parents and caring for an unwell patient (infant), priority of care, communications, adequately prepare families<br>Transitioning support across the continuum<br>Empowering families to feel confident, parent involvement, support for the family, fluid relationship that changes with needs, empowerment, acting as role models<br>Definition: Not provided. Places importance on the application of family centred care and the need for ongoing organisational support, guidance and education. | |
| Edvardsson, Varrailhon, Edvardsson (2014) Sweden | Swedish nurses in aged, long term care N = 436 | Anthropological free listing and qualitative content analysis | Promoting decision making<br>Involvement in decision making, activities, offering choice, respecting preferences, understanding patient history, respecting residents view point, respecting lifestyle choices and routines<br>Promoting meaningful life<br>Individually targeted life, involvement in everyday life tasks, creating activities, sharing knowledge, playing games, listening, having appropriate space and activities<br>Promoting a pleasurable living<br>Little extras, experience, pleasure, being careful<br>Promoting personhood<br>Life stories and meaningful interaction, seeing the resident as valuable, making eye contact, using name, greeting and acknowledging the person, sharing a meal or a coffee<br>Asking questions and taking interest<br>Definition: Defines PCC as a philosophy of care and a culture rather than singular interventions. | |

(*Continued*)

**Table 2.** (Continued)

| Authors, year, location | Population | Design methods | Findings/Comments | Common authors and articles |
|---|---|---|---|---|
| Ross, Tod, Clarke (2014) UK | Nurse perspectives of PCC Acute medical ward Semi structured interviews N = 14 | Semi Structured interviews and thematic analysis | Characteristics of relationships<br>Being friendly and approachable, finding the time to listen and talk with patients, freely give information and ease anxiety<br>Team contributions<br>Personal qualities of staff<br>Relationships are valued, respect for beliefs, being recognised as important, patient stories, flexibility in ward routine, role modelling<br>Respecting the principles of PCC<br>Recognising the importance of personal wishes and values and being considerate of decisions, supporting the person to ask questions, being an advocate, responsive assessments, dignity<br>Definition: Characteristics of relationships, personal qualities of staff and respecting the principles of PCC act together to shape the philosophy of care in the medical ward. In turn the philosophy influences how staff perceive and facilitate PCC. | |
| Bala, Forslind, Fridlund, Samuelson, Svensson, Hagell (2017) Sweden | Outpatient rheumatology clinics N = 50 | Qualitative questionnaire and thematic analysis | Social Environment<br>Approached and communicated with, good relationships, establishing a warm calm friendship, creating an environment which limits disruptions, engaging and safe<br>Personalisation<br>Preferences and values of the individual, communication, planning is tailored and collaborative, a space to tell stories and use personal information<br>Shared decision making<br>Collaborative and interpersonal, discussion about options, free choice and respect for choice, sharing information with other care planners and planning follow up<br>Empowering<br>Individuals resources and abilities are considered, support, mutual process, power autonomy and responsibility<br>Active role in care, provided with opportunities, strengthening self-confidence<br>Listening, encouraging<br>Definition: Not provided, however provides a conceptual framework, therefore contributes to meaning | |

## Theme 2: Practice

PCC is a product of person-centred practice, particularly in the context of nursing. However, the ability to practice PCC is influenced by professional and system factors. The core theme of Practice is comprised of the sub-themes *Doing* and *Space*.

**Doing.** 'Doing' refers to the complex interplay of professional attributes, behaviours and tasks that makes up the daily remit of the nurse; that is, the 'doing' of nursing is a combination of these things within the care environment. Personal attributes of nursing staff emerge as a common element in the literature related to PCC. The literature describes attributes such as communication, respect, values, empathy, compassion and non-judgemental behaviour [9,29]. Lusk and Fater [7] further describe such attributes as caring, faith and hope, trust, relationships, teaching, learning and listening. In describing a framework to facilitate the practice of person-centred care, McCormack and McCance [22] discussed professional competence, interpersonal skill, job commitment and professional insight. Others have extended this to include understanding vulnerability, fear, the patient identity [30] and highlight the need to recognise the person as competent to make decisions [31]. This view, centred on dignity and privacy and the moral and ethical behaviours of the nurse [16,28], facilitates the relationship and balance of power with the person. Delivering whole person care includes elements such as

**Table 3. Summary of integrative review themes.**

| Theme | Subtheme | Findings |
|---|---|---|
| People | Recognising Uniqueness | • The person is an individual with unique needs from the healthcare team<br>• The need to develop and maintain trust<br>• Care is tailored for the person |
| | Partnership | • A professional relationship is formed<br>• Mutuality between provider and receiver is fostered |
| Practice | Doing | • Professional attributes of the nurse such as respect, compassion and non-judgemental behaviour<br>• Professional competence in practice<br>• Recognising the person as competent to make decisions and valuing the lived experience of the person<br>• Meeting the physical and emotional needs of the person |
| | Space | • Being flexible and offering choice while creating opportunities for the person to engage<br>• Freely giving information and finding time to listen and engage<br>• Opening the space for PCC despite competing priorities<br>• Positioning PCC as the major priority in care |
| Power | Power Over One's Care | • Balancing the power between the provider and the receiver; A marriage between provider and receiver allowing for a sharing of knowledge<br>• Fostering autonomy and participation |
| | The Power to Practice PCC | • Healthcare systems and environments must be conducive to PCC<br>• Systemic barriers such as workplace culture, leadership, policy must be addressed<br>• The measurement of PCC in its current form may be more system centred, hence perpetuating task orientation among other things |

respect for the individual [16], and planning care that is based on individual needs [25]. In practice, this is described as the person being valued for their lived experience, life stories [9,16] and the continuation of self [34]. Kitson et al. [35] describe this as addressing both the physical and emotional needs of the person and alleviating anxiety.

**Space.** The existing literature alludes to the idea of opening a space to practice PCC. The literature describes this as being flexible within the care, offering choice [9,31], and creating opportunities for people to engage [34]. Practicing PCC involves freely giving information to the person [36] and finding the time to listen and engage with them [36,39], which implies that PCC is a proactive way of delivering nursing care. Gachoud et al. [32] found that nurses see themselves as most important in delivering PCC, with Doctors playing a lesser role in PCC practice. This understanding creates a concept whereby nurses are pivotal in creating an environment in which the person can truly engage.

While PCC is an individual practice method, the environment within which nurses' practice must be considerate and supportive of the delivery of PCC as a significant priority; a view supported by McCormack and McCance [22] in their description of organisational systems and leadership within PCC. Despite competing priorities and the associated tasks of daily practice, nurses must find and open a space to practice PCC as an essential element of the profession. Interestingly, PCC within the literature is often discussed as an addition to nursing tasks. Edvardsson et al. state that promoting PCC in aged care includes doing 'little extras' [9 p50], such as understanding the patient's life story, making eye contact and using the person's name. Marshall et al. found that nurses describe PCC as 'making the effort' and 'going the extra mile'

[37 p2667], being helpful and timely with care and attention. Others describe making choices available [16], ascertaining priorities [27] and doing the 'right' thing [31].

### Theme 3: Power

PCC as a concept is about balancing power between the provider and receiver of care. The notion of PCC is imbued with connotation of power, discussed in relation to all elements of the care and is intertwined in some way with all themes within this review. The sub-themes of Power are the *Power over one's care* and the *Power to practice PCC*.

**Power over one's care.** The idea of power balance is discussed in the literature and includes the sharing of knowledge [29], respect for decision making and individualised care based on these decisions [28,17]. Further to this is the notion of the person having 'active' involvement in the care process [22,35]. This is described through identification of the person's strengths and reinforcing this through the care continuum [16,38]. In addition to this, the literature describes empowerment, promoting the sense of self efficacy [31,33,37], supporting the person to be as self-managing as possible [36] or to have a level of autonomy in their care. Here, the person holds the power in care planning and decision making throughout the care journey and there is a responsibility of knowledge transference and the maintenance of personal autonomy [35]. This is apparent in the literature through concepts such as control, rights, patient involvement and participation [27,33,35]. PCC, however, places importance on a marriage between provider and receiver as a process of sharing knowledge, rather being entirely self-governing, in which the provider (as the custodian of knowledge) has an obligation to impart knowledge.

**The power to practice PCC.** The need for care systems to be innovative and make a commitment to PCC comes through in the literature [22], as well as the need for the environment to allow for flexibility and to factor time and space to practice PCC [9,36]. This is a significant shift from the traditional biomedical model, whereby emphasis on personal choice [33] and partnerships [17] must be considered within all layers of the healthcare system. Barriers and enablers including workplace culture, leadership [22], policy and practice, organisational systems, environment [28,35], workload, and ward culture [37] were identified. The literature also included topics around cost [28], care coordination [28,33] and of course, outcomes of clinical care provided [27,29,30].

Jakimowicz et al. [30] noted the conflict between system standards, benchmarking and the provision of PCC in a time poor environment. Consistently, the literature discussed the idea of measuring PCC as a method of quantifying this important element of nursing practice amongst the myriad of measurable tasks nursing time is allocated to. The need and ability to measure PCC is cited as crucial for quality improvement of care [31, 38]. This review excluded articles related to the measurement of PCC as the primary aim was to find how PCC was defined, however this was still very much a part of the discussion around the meaning and practice of PCC. Morgan and Yoder [31] discussed several measurement tools, finding them to align more with the *effect* of care rather than the care directly. Lawrence & Kinn [27] found that outcome measures used where often in line with the needs and requirements of clinicians, auditors and researchers, or hospital clinical outcomes [33], rather than with the goals of the patient. Outcomes vary from self-care, patient satisfaction, well-being and improved quality of care [28,31,38] to improved adherence and decreased hospitalisation [29]. This highlights competing priorities within the nursing profession and demonstrates that nursing time is conflicted between what they 'should' do and what they 'must' do, hence highlighting a nurses limited power to practice PCC in the context of the system standards.

## Discussion

The review demonstrates that the concept of PCC is indeed a method of providing care, or the way in which nurses deliver care. To be person-centred, the nurse must recognise the person as unique, form meaningful partnerships, open a space within the doing of their day to involve and engage with the person, allowing the person control and power of their care.

It is interesting to note that while the existing literature covers a wide variety of clinical areas, and patient and staff perspectives, there were indeed core common themes of PCC. Despite the core concepts associated with PCC taking on more importance within certain clinical areas; for example, continuation of self in aged care [9,34], patient advocacy for intensive care [30], or communication in stroke care [27], they are consistent across specialities with the themes building on one another. Perhaps the reason why PCC has been so widely accepted is that the characteristics are simple, kind, human interactions, valuing both the person and the care provider. While definitions of PCC exist, there is no one universally used definition of PCC in nursing practice, potentially compounding a degree of separation between practice and healthcare systems. The findings demonstrate a tension between the theory and the conceptualisation of PCC, and as a result, the operationalisation of the term at both the practice level and a wider healthcare service level.

At the practice level, the theory/practice gap for PCC was evident. The theory/practice gap includes elements of practice failing to reflect theory, perceptions of theory being irrelevant to practice, and ritualistic nursing practice. Consequences of the theory/practice gap can greatly influence nursing practice and collaboration [40]. In the context of PCC, the theory/practice gap is apparent in the challenge of translating the ideas of PCC into a concrete concept. It is of significance that PCC is seen as 'extra' or additional to nursing tasks when these professional behaviours are in line with the Australian Nursing Professional Standards [8], which requires that they are an intrinsic element of the nursing profession. In fact, to be a registered nurse in Australia one must demonstrate respect for the person as the expert, respect autonomy and "*share knowledge and practice that supports person-centred care*" [8]. This highlights an important matter for consideration; why are core elements of PCC being viewed as 'going the extra mile' rather than a core competency for nurses? Certainly, from the perspective of the professional standards, PCC should not be the road less travelled, but rather the daily standard practice of nursing. One answer to this may be the task orientation of the contemporary nursing culture that sees nurses required to meet organisational time allocations for care [41]. Sharp, McAllister and Broadbent [42] uncovered a tension between PCC and nursing culture, finding that nurses were increasingly bogged down with tasks and processes, taking them away from the people that they provide nursing care for. These authors found that this led to a feeling of frustration and helplessness in nurses who appear to have accepted the culture of auditable, measurable activities and processes, particularly within the climate of organisational accreditation requirements. This activity-based nursing environment manifests in missed nursing care largely related to patient centred elements, e.g. discharge planning, communication within the healthcare team, absence of adequate patient education on key factors of care such as medication guidance, functional assessment and so on [43,44].

Further, it is apparent that the concept of PCC cannot be isolated from other philosophies of nursing practice and in fact, is embedded in other approaches to nursing care. For example, as outlined by Kim [45], nursing is defined by dimensions, rather than characteristics. If PCC is considered as a dimension, a complex, interwoven mix of characteristics, then it is possible to gain some concrete understanding of PCC in the context of all clinical areas. The five dimensions proposed by Kim reflect the 'human' side of nursing practice, and like the general interpretation of PCC, shows how human interactions, values and knowledge combine to

provide care. Kim goes on to say that these dimensions vary with individual nurses and changing clinical situations; which seems to fit with the current confusion about PCC giving choice and decision-making power to patients. As found in this review, PCC attempts to balance the power between providers of care and receivers, giving choice and decision-making power. Yet the focus on nursing tasks and prioritisation of these tasks is evident, demonstrating the malalignment between concept and practice, where research has identified that the current task-oriented system of nursing does fail to meet the care needs of patients [46].

Nursing practice, however, is only one element of delivering PCC within the healthcare system. This disparity extends between the concept of PCC and its ability to exist within the current healthcare system itself, where time to care is explicitly rationed through budgets that do not allow for individualised person-centred care [47].

The notion of PCC is one centred on mutuality and a balance of power; a distinct move from the paternalistic biomedical model to a biopsychosocial model that is guided by the person, rather than the disease process. However, in current healthcare services, care is often system centred. That is, care is organised, funded and coordinated in a way that meets the needs of the system or service [28,33]. System fragmentation is understood to have significant influence on people accessing care, whereby people with long-term and complex conditions are most vulnerable to the negative impact from the lack of care coordination and cohesion [48].

In Australia, complex funding models are central to the concept of system fragmentation which begins at the Commonwealth and State funding levels [48], making it difficult for patients to navigate the system. System silos remain a significant issue for healthcare services and for the delivery of care, with Medicare models remaining fragmented for specialist services [49]. The States are the healthcare system managers, yet the federal government holds the responsibility of leading primary healthcare. This presents a challenge in provided collaborative and integrated services, particularly for those with long-term conditions [50]. The OECD highlight the importance of reducing system fragmentation in order to 'Improve the co-ordination of patient care.' [50 p1].

Indeed, system fragmentation leads to an increased 'treatment burden', whereby poor treatment coordination, ineffective communication and confusion about treatments can contribute to poor health outcomes and greater levels of cost, time, travel and medications for the person [51] Sav et al. [51] discuss the need for individualised and coordinated services across specialities as a requirement for reducing treatment burden. In addition to this, the Australian Charter of Healthcare Rights prescribes the rights of those seeking care in any Australian service and includes the right to access, respect, communication and participation [52]. Accreditation of healthcare services is conditional to evidence of multi-level partnerships with consumers of health. Positive partnerships (PCC) are clearly linked to improved access to care, which in turn leads to reports of positive experiences and better-quality healthcare. Of critical importance at an organisational and government level, the standards also describe this partnership as a mechanism for reducing hospital costs through improved rates of preventable hospitalisation and reducing hospital length of stay [7].

Potentially Preventable Hospitalisations (PPH) place considerable economic and resource burden on the healthcare system, with approximately 47% of PPH being attributed to long-term conditions [53]. Thus, reducing preventable hospitalisation is a measurable target for healthcare services under the National Healthcare Agreement as a way of controlling the escalating costs of care and maintaining sound fiscal management of public services [54]. In line with the ACSQHC standard *Partnering with Consumers*, PCC has been introduced to some services as a mechanism for improving communication between services and those with long term conditions. What is less clear, is how care tailored to the individuals wants and needs of the patient (PCC), exists within a system predominately focused on reducing variation and the

associated costs of care. While the philosophy of PCC naturally fits within the care environment, understanding how effective it is, how the person is included and how outcomes important to the person are captured, take a lower precedent to the measure of reduced hospital costs, self-efficacy and reduced hospitalisation. Capturing what is important for the individual presents a difficult task for services providing population-based care.

These system wide constraints provide a considerable challenge to nurses in their attempts to operationalise the concept of PCC. Nursing, it seems, has become task orientated, a sentiment supported by Foe & Kitson who found that nurses are constrained by a 'checklist' mentality, whereby completing and documenting tasks is seen as more important than engaging with the person [55 p100]. These tasks and checklists align with the requirements of the National Standards for hospital accreditation. An example of this is the need to collect data on the use of invasive devices or the allocated time intervals in which screening (such as skin inspection and falls risk) must occur; for example within eight (8) hours of admission [7]. Indeed, policy and procedure for nursing practice reflect that of the need of accreditation and national policy requirements as opposed to the needs of individual people. While partnering with consumers is an important element of the standards [7], quantifying the way in which healthcare services and indeed nurses engage with patients is less clear. Kitson states '*Nursing theory, it would seem, has been limited by the profession's ability to systematically document the complexity and richness of what happens when nurses and patients (and their careers) interact*' [35 p99], an issue it seems stemming from the fact that nursing interventions promoting person-centred, compassionate care are poorly described, with little to no consensus on the term, and interventions that do exist are poorly evaluated [56]. On top of this, nurses are generally not encouraged, nor enabled to reflect on practice in order to generate new insights and nursing practice [35]. Molina-Mula et al. [23], discuss the nursing profession as being the key to professional teamwork models, meeting the needs of patients and thereby increasing their personal decision-making capacity. However, it is possible that PCC is hindered by the level of professional autonomy, time and space afforded to nurses [57]. Indeed, the malalignment discussed herein, demonstrates that nurses may be hindered at higher levels of system compliance or difficulties in coordinating care services, which permeates nursing culture and ultimately nursing practice, limiting their ability to provide PCC that is individualised to the people seeking care.

Finally, while this review excluded articles related to the measurement and indicators of PCC, this is undoubtedly linked to its perceived meaning and how it is operationalised. This review demonstrates that the understanding of PCC is made up of how and where PCC appears in healthcare discourse and shows that PCC is potentially skewed by how it is(n't) measured and the outcomes that are(n't) reported as a product of PCC. This finding presents a framework within which further investigation of the concept of PCC (Meaning, Practice, Measures, Outcomes) within healthcare services could be undertaken. This proposed framework will be applied by the author to conduct further research into the role of PCC within nurse-led service for people with long-term conditions.

## Implications for practice

This review highlights the dominant discourse around the concept of PCC yet uncovered the idea of malalignment between the rhetoric and the reality of the concept. Further exploration of the alignment between healthcare services and the goal of PCC may prove beneficial in ensuring the practice of PCC is fostered from all levels of the healthcare service. The above provides a rationale for why the definition of PCC should be provided, given that the concept is currently somewhat nebulous in nature. A consistent definition, with reference to all levels of the healthcare service including practice, will ensure that the concept stays true to the

philosophy of compassionate and balanced care. Any definition provided should carefully consider how PCC is measured and prioritised within the healthcare system, which has the potential to move the concept from its current rhetorical nature, to a genuine commitment and priority of nurses and services. Lastly, the review provides a basis for the importance of nursing education and workforce development of the concept and the practice of PCC, given the apparent barriers that nurses may face in delivering PCC.

## Limitations

This integrated review was limited to articles relating to the nursing profession and hence has excluded reviews on PCC in relation to other disciplines. Practice related elements such as procedure, service measures and outcomes of PCC were excluded from this review as the aim was to find a generalised way of defining the term. Furthermore, only one framework met the criteria of the search strategy and was included, there are however, several frameworks for PCC in nursing and hence some elements of PCC and their definitions may have been excluded.

This review was performed with published literature only, with no investigation of grey literature undertaken. PCC is often discussed in healthcare service literature, including procedure, service profiles and service strategy. This information will undoubtedly have an impact on how nurses understand and practice PCC within their own area and within their service. This review was designed to investigate a universal definition of PCC as described in the literature and hence chose to limit this to an academic search. The practice of PCC from a policy to practice perspective perpetuates meaning and will be the subject of further research for the author.

This review was conducted as a starting point for the author's research higher degree (PhD) studies, and hence the search strategy and quality processes were completed by one person. All elements of the review were discussed at length with academic supervisors to ensure adequate rigor and accuracy throughout the search, review and integrative process.

## Conclusion

The concept of PCC is well known to nurses, yet ill-defined and operationalised into practice. Healthcare service policy and care provisions, and indeed nursing services, need a clear definition of PCC in order to work toward embedding it into practice and into models of care in a meaningful and genuine way. However, PCC is potentially hindered by its apparent rhetorical nature, and further investigation of how PCC is valued and operationalised through its measurement and reported outcomes will serve the philosophy of PCC well. Investigation of the literature found many definitions of PCC, but no one universally accepted and used definition. Subsequently, PCC remains conceptional in nature, leading to disparity between how it is(n't) operationalised within the healthcare system and within nursing services. In light of the malalignment discovered within this review, a universal definition of PCC is not provided herein; instead, this review highlights the need for further investigation of PCC between the levels of the healthcare service (at the micro, meso and macro levels) and how this influences the critical work that nurses do in supporting people through their healthcare journey.

## Supporting information

**S1 File. PRISMA checklist.**
(DOC)

## Author Contributions

**Conceptualization:** Amy-Louise Byrne, Adele Baldwin, Clare Harvey.

**Data curation:** Amy-Louise Byrne.

**Formal analysis:** Amy-Louise Byrne.

**Investigation:** Amy-Louise Byrne, Clare Harvey.

**Methodology:** Amy-Louise Byrne, Adele Baldwin, Clare Harvey.

**Project administration:** Amy-Louise Byrne.

**Resources:** Amy-Louise Byrne.

**Supervision:** Adele Baldwin, Clare Harvey.

**Validation:** Adele Baldwin, Clare Harvey.

**Visualization:** Amy-Louise Byrne, Adele Baldwin.

**Writing – original draft:** Amy-Louise Byrne.

**Writing – review & editing:** Amy-Louise Byrne, Adele Baldwin, Clare Harvey.

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
