## [Decision Letter · Decision Letter 0]

2 Jan 2020

PONE-D-19-26103

Whose Centre is it Anyway? Defining Person-Centred Care in Nursing

PLOS ONE

Dear Mrs Byrne,

Thank you for submitting your manuscript to PLOS ONE. After careful consideration, we feel that it has merit but does not fully meet PLOS ONE’s publication criteria as it currently stands. Therefore, we invite you to submit a revised version of the manuscript that addresses the points raised during the review process.

We would appreciate receiving your revised manuscript by Feb 16 2020 11:59PM. To enhance the reproducibility of your results, we recommend that if applicable you deposit your laboratory protocols in protocols.io, where a protocol can be assigned its own identifier (DOI) such that it can be cited independently in the future. For instructions see: http://journals.plos.org/plosone/s/submission-guidelines#loc-laboratory-protocols

We look forward to receiving your revised manuscript.

Kind regards,

Janhavi Ajit Vaingankar

Academic Editor

PLOS ONE

Journal Requirements:

2. Please include the study type, Integrative Review, in the title. Please cite articles where those Integrative Review protocols are described in detail, and include sufficient information the methods to be understood independent of these references (https://journals.plos.org/plosone/s/submission-guidelines#loc-materials-and-methods).

3. We note you have included a table to which you do not refer in the text of your manuscript. Please ensure that you refer to Table 2 in your text; if accepted, production will need this reference to link the reader to the Table.

4. Please ensure that you refer to Figure 1 in your text as, if accepted, production will need this reference to link the reader to the figure.

Reviewers' comments:

Reviewer's Responses to Questions

**Comments to the Author**

1. Is the manuscript technically sound, and do the data support the conclusions?

Reviewer #1: Yes

Reviewer #2: Yes

2. Has the statistical analysis been performed appropriately and rigorously? 

Reviewer #1: Yes

Reviewer #2: N/A

3. Have the authors made all data underlying the findings in their manuscript fully available?

Reviewer #1: Yes

Reviewer #2: Yes

4. Is the manuscript presented in an intelligible fashion and written in standard English?

Reviewer #1: Yes

Reviewer #2: Yes

5. Review Comments to the Author

Reviewer #1: Praise to the authors for having written a good review highlighting seminal information on Person-Centered Care (PCC). The methodological approach is clear and appropriate. The approach used in the process of eligibility of studies included is also clear with a supporting table, however, it is not clear how quality appraisal using CASP was applied to the studies included.

Results: The results are presented concisely, clearly and make sense with respect to the aim of the review but could have been longer to elaborate on the characteristics of the papers included in the review.

Discussion: The discussion is good and linked to theory and practice. However, a higher level of discussion was needed here. There could have been a better debate highlighting the linkage or differentiating features of PCC from similar concepts such as patient centered care or patient and family centered care.

Conclusion: Very clear and linked to the study findings. The fact that authors attempted to highlight current literature on person centered care, this could have been more interesting if they could have highlighted the differences in terms of definition by the different scholars or institutions.

Advice for improvement of the manuscript: as this is an integrative review, it could offer more in the way of future direction for defining the concept of PCC.

Reviewer #2: This is an interesting paper that has the potential to inform policy change in the future. The review itself seems quite strong, however, the international/Australian focus needs more clarity. In the literature search strategy you have chosen to include multiple jurisdictions which have important differences in healthcare system functioning. However, in each section when you give examples they are only focused on the Australian context. While I understand that for the definition of PCC the inclusion of more broad context is helpful, consider carefully how the operationalization will be different in different jurisdictions and the policy/practice challenges. I suggest you either make it clear that this global review of definitions is only being applied to the Australian context, or diversify your examples throughout. This will also be important for the corresponding terminology that you use – the term consumer for example is rarely used in the Canadian healthcare context so it will be important early in the paper to define such terms and outline which contexts they are used and comparable terms in each of the other jurisdictions. Also related to terminology in Canada, we would refer only to the ‘healthcare system’, not the ‘health system’ as there is an acknowledgement that it is not about preventative health.

Here are some more minor changes throughout:

Introduction:

Page 3, line 47-49 “The concept of Person-Centred Care (PCC) is used to describe the role of the patient within the health system and the way in which care is provided to the patient [2,3].”

- This is to describe a certain model for the role of the patient – not the role of patients in general – this implies it is a way to describe patients in all care models

Page 3, line 52-58 – accreditation is based on jurisdiction – which jurisdiction are you referring to? Same with nursing professional standards.

Page 4, line 59 – healthcare not health

Page 4, line 63 – is this different nursing contexts as in hospitals vs community or different countries where they practice?

Background:

Page 5, line 84 – when was this report created? What happened between 1977 and 2000? Add a sentence here, likely wasn’t a 20-30 year period without any changes.

Page 5, line 86 – is this the first definition of PCC? It’s the first you mention as far as I can tell for this specific term, might be important to mention this if it is the first attempt to define it.

Line 89 – is this WHO framework? Unclear who’s framework – please specify

Line 94 – awkward “of their own care” consider changing to about their own care or over their own care.

Line 96-97 – this quote is cut a little awkwardly – consider adding a word in square brackets to make it flow better like [involving].

Line 98 – from not form

Again your aim and scope doesn’t specify an Australian context, rather an international application. Either make clear in the introduction that this paragraph about Australia’s ACSQHC is an example from the various jurisdictions. Potentially consider including an example from another country to support the international focus of the paper.

Paragraph starting line 104 seems disconnected from previous paragraph. Consider adding a sentence at the beginning that bridges and waiting until the second sentence to list examples of frameworks. Consider moving the sentence on 110 to this paragraph as it justifies the move from PCC in general to nursing frameworks specifically.

Line 114-116 – combine these 2 short sentences.

Aim:

- The first aim – to better understand the literature – seems vague, can you be more specific here? Consider combining with the search question – ie the aim could be to understand better from the literature how nurses operationalize the definition.

Methods:

Line 144 – you don’t specify you use this framework, make this clear

Figure 1 – are you missing the arrow from screening to eligibility?

How many papers are citing the same papers?

Findings:

Consider creating a figure or table summarizing each of the themes and subthemes to allow a visualization of your results that is easier for readers to refer to.

Line 200 – ‘based’ not ‘base’.

Line 237 – take out the “the” before people.

Line 240 – need to define the term medical officer, either in this paragraph or as a footnote.

Line 281 – access to what?

Line 290 – if measuring PCC is important to the discussion and still part of the papers you reviewed it would be helpful to add a sentence or two here to justify why you didn’t include them.

Discussion:

Lines 302-305 – abrupt switch between these two sentences, consider a sentence in between moving from stating it’s a model of care to what the reader must assume is the definition that emerged from the review.

Line 325 – they are ‘an’ intrinsic element.

Line 330 – core business – consider core competency or another word that is more in line with healthcare and patient centredness.

Line 354 – this sentence seems unfinished – should it say “giving choice and decision-making power to patients”?

Line 360 – again should be healthcare system throughout.

Again this is an Australian example, consider diversifying or adding a second country example to make the argument that your paper has international scope.

Implications for practice:

Again this is another section to clarify the jurisdiction you are talking about and be careful about the suggestions you are making here and in the conclusion about policy and practice. It could be helpful to say that across the western countries included general things need to change and give a couple of specific examples in different countries.

6. PLOS authors have the option to publish the peer review history of their article (what does this mean?). If published, this will include your full peer review and any attached files.

Reviewer #1: Yes: Frank Kiwanuka

Reviewer #2: No

---

## [Author Response · Author response to Decision Letter 0]

9 Jan 2020

Thank you for the comprehensive feedback on the manuscript Whose Centre is it Anyway? Defining Person-Centred Care in Nursing; An Integrative Review. The valuable feedback has been considered and changes have been made as per the table below. Please find attached a revised copy of the manuscript with tracked changes and a complete copy with changes embedded. Once again, we thank you for your time and consideration and we look forward to your consideration of this new version.

Yours sincerely

Amy-Louise Byrne (Corresponding author)

Please see the rebuttal letter for a detail account of the changes made

---

## [Decision Letter · Decision Letter 1]

4 Feb 2020

PONE-D-19-26103R1

Whose Centre is it Anyway? Defining Person-Centred Care in Nursing: An Integrative Review

PLOS ONE

Dear Mrs Byrne,

Thank you for submitting your manuscript to PLOS ONE. After careful consideration, we feel that it has merit but does not fully meet PLOS ONE’s publication criteria as it currently stands. Therefore, we invite you to submit a revised version of the manuscript that addresses the points raised during the review process.

We would appreciate receiving your revised manuscript by Mar 20 2020 11:59PM. To enhance the reproducibility of your results, we recommend that if applicable you deposit your laboratory protocols in protocols.io, where a protocol can be assigned its own identifier (DOI) such that it can be cited independently in the future. For instructions see: http://journals.plos.org/plosone/s/submission-guidelines#loc-laboratory-protocols

We look forward to receiving your revised manuscript.

Kind regards,

Janhavi Ajit Vaingankar

Academic Editor

PLOS ONE

Reviewers' comments:

Reviewer's Responses to Questions

**Comments to the Author**

1. If the authors have adequately addressed your comments raised in a previous round of review and you feel that this manuscript is now acceptable for publication, you may indicate that here to bypass the “Comments to the Author” section, enter your conflict of interest statement in the “Confidential to Editor” section, and submit your "Accept" recommendation.

Reviewer #1: (No Response)

2. Is the manuscript technically sound, and do the data support the conclusions?

Reviewer #1: Yes

3. Has the statistical analysis been performed appropriately and rigorously? 

Reviewer #1: N/A

4. Have the authors made all data underlying the findings in their manuscript fully available?

Reviewer #1: Yes

5. Is the manuscript presented in an intelligible fashion and written in standard English?

Reviewer #1: Yes

6. Review Comments to the Author

Reviewer #1: The authors have sufficiently responded to the previous comments and thanks for submitting the revision. However, they could consider expanding on the background by including more literature from other countries which have significantly contributed to literature person-centered care. Some suggestions, there is plenty of literature coming from the University of Gothenburg Center or Person centers care GPCC and Finland on this topic. This will reflect a global picture on the topic. Were there any assumptions that the authors made about including the countries specified?

7. PLOS authors have the option to publish the peer review history of their article (what does this mean?). If published, this will include your full peer review and any attached files.

Reviewer #1: Yes: Frank Kiwanuka

---

## [Author Response · Author response to Decision Letter 1]

5 Feb 2020

The feedback from the reviewers is appreciated. Please see the rebuttal letter for a response to the reviewers. No changes have been made to the manuscript on this third submission

---

## [Editor Report · Decision Letter 2]

19 Feb 2020

Whose Centre is it Anyway? Defining Person-Centred Care in Nursing: An Integrative Review

PONE-D-19-26103R2

Dear Dr. Byrne,

We are pleased to inform you that your manuscript has been judged scientifically suitable for publication and will be formally accepted for publication once it complies with all outstanding technical requirements.

With kind regards,

Janhavi Ajit Vaingankar

Academic Editor

PLOS ONE
---

## [Editor Report · Acceptance letter]

25 Feb 2020

PONE-D-19-26103R2 

Whose Centre is it Anyway? Defining Person-Centred Care in Nursing: An Integrative Review 

Dear Dr. Byrne:

I am pleased to inform you that your manuscript has been deemed suitable for publication in PLOS ONE. Congratulations! Your manuscript is now with our production department. 

With kind regards,

on behalf of

Ms Janhavi Ajit Vaingankar 

Academic Editor

PLOS ONE